# Sex-dependent noradrenergic modulation of premotor cortex during decision-making

Ellen M Rodberg, Carolina R den Hartog, Emma S Dauster, Elena M Vazey*

Neuroscience and Behavior Program and Department of Biology, University of Massachusetts Amherst, Amherst, United States

**Abstract** Rodent premotor cortex (M2) integrates information from sensory and cognitive networks for action planning during goal-directed decision-making. M2 function is regulated by cortical inputs and ascending neuromodulators, including norepinephrine (NE) released from the locus coeruleus (LC). LC-NE has been shown to modulate the signal-to-noise ratio of neural representations in target cortical regions, increasing the salience of relevant stimuli. Using rats performing a two-alternative forced choice task after administration of a β-noradrenergic antagonist (propranolol), we show that β-noradrenergic signaling is necessary for effective action plan signals in anterior M2. Loss of β-noradrenergic signaling results in failure to suppress irrelevant action plans in anterior M2 disrupting decoding of cue-related information, delaying decision times, and increasing trial omissions, particularly in females. Furthermore, we identify a potential mechanism for the sex bias in behavioral and neural changes after propranolol administration via differential expression of β2 noradrenergic receptor RNA across sexes in anterior M2, particularly on local inhibitory neurons. Overall, we show a critical role for β-noradrenergic signaling in anterior M2 during decision-making by suppressing irrelevant information to enable efficient action planning and decision-making.

*For correspondence:
evazey@umass.edu

Competing interest: The authors declare that no competing interests exist.

## Editor's evaluation

This study provides valuable evidence of a potential sex-dependent, biological mechanism for propranolol-driven changes in decision-making. Though the functional component directly linking premotor cortex activity to performance is incomplete, the discovery of a molecular bias in adreno-receptor expression that may underlie its impact on behavior is novel and compelling. These experiments present a useful framework for future inquiry linking direct actions within premotor cortex for decision-making changes in both males and females.

## Introduction

Successfully navigating through an environment requires internally guided behaviors and the use of external cues to make decisions and drive goal-directed actions. Decision-making is a complex and multifaceted process, many aspects of which can be interrogated in rodents. Critical components of the decision process include detecting and encoding stimuli, forming and maintaining action plans, preparing and executing an action, and comparing predicted outcomes to the actual results of that decision. Action planning is the neural representation of an intended movement and can be internally generated or externally guided by relevant cues. When externally guided, a neural action plan emerges after cue onset that persists until the action is executed. One key brain region with strong action plan representation is secondary motor cortex (M2) in rodents, analogous to Brodmann's area 6, supplementary motor area (SMA), and pre-SMA in primates. M2 is interconnected with both sensory,

motor, and cognitive brain regions (*Hoover and Vertes, 2007*; *Jeong et al., 2016*; *Reep et al., 1990*; *Reep and Corwin, 1999*) and integrates information from these networks to facilitate planning, initiation, and execution of internally generated or cue-guided goal-direction actions (*Barthas and Kwan, 2017*; *Gremel and Costa, 2013*; *Inagaki et al., 2018*; *Sul et al., 2011*; *Wei et al., 2019*). Neurons in M2 display heterogeneous activity during decision-making and individual neurons are tuned to differentially encode distinct action plans with either excitation or active inhibition (*Chandrasekaran et al., 2017*; *Inagaki et al., 2018*; *Wei et al., 2019*). M2 neurons display some of the earliest choice-related activity in the fronto-striatal network (*Sul et al., 2011*), encoding goal-directed action plans with neural signatures such as cue-evoked firing, activity ramping, and sustained activation during delays (*Inagaki et al., 2018*; *Murakami et al., 2014*; *Svoboda and Li, 2018*). The separation of opposing action plans in M2 is necessary for accuracy on cue-guided tasks; direct or indirect bilateral inhibition of select action plans in M2 impairs performance on cue-guided tasks (*Hanks et al., 2015*; *Li et al., 2016*). Furthermore, targeted photostimulation of select action plans is sufficient to drive behavioral responses (*Daie et al., 2021*).

Investigating what influences action plan activity in M2 is critical to understand the mechanics of goal-directed actions. In addition to cognitive and motor network connections, M2 is innervated by neuromodulators including norepinephrine (NE) from the locus coeruleus (LC) in the brainstem (*Agster et al., 2013*; *Hoover and Vertes, 2007*). The LC is a small cluster of NE containing neurons that project throughout the brain and are the near exclusive source of NE to the cortex, including M2 (*Aston-Jones, 2004*; *Moore and Bloom, 1979*). Causal studies have shown a direct role for phasic NE in increasing gain for task-relevant stimuli (*Clayton et al., 2004*; *Dayan and Yu, 2006*; *Servan-Schreiber et al., 1990*; *Vazey et al., 2018*; *Waterhouse et al., 1998*). In this way, LC-NE facilitates decision-making by increasing the salience of task-related cues and maintaining attention. Although the anatomical connectivity between LC and M2 is well documented, whether and how NE facilitates action planning in M2 is unclear.

Importantly, there is limited evidence on whether the function of noradrenergic modulation on action planning in M2 is similar across females and males. There is strong potential for interaction in these domains as there are known sex differences in both neuromodulatory clusters that provide input to M2, including LC (*Bangasser et al., 2011*; *Pinos et al., 2001*; *Valentino et al., 2012*) and behavioral strategies for decision-making (*Chen et al., 2021a*; *Chen et al., 2021b*; *Orsini and Setlow, 2017*; *Shansky, 2018*).

Here we investigated how noradrenergic tone regulates anterior M2 function and action planning in female and male rats. We recorded neural activity from anterior M2 and modulated noradrenergic signaling through β-noradrenergic receptors during a simple decision-making task, two-alternative forced-choice (2AFC). We found noradrenergic blockade decreased behavioral performance and dampened the neural representations of action plans in anterior M2, specifically by decreasing the active inhibition of irrelevant action plans. Our results show the impact of noradrenergic blockade on performance and neural activity was greater in females than males. To identify a potential mechanism for these differences in β-noradrenergic sensitivity, we characterized β1- and β2-noradrenergic receptor RNA levels in M2 of male and female rats. We found increased β2-noradrenergic receptor RNA expression in females that may underlie the sex-specific sensitivity to noradrenergic regulation of action planning.

## Results

### β-Noradrenergic inhibition decreases engagement and trial completion in 2AFC

We sought to examine the role of norepinephrine, specifically β-noradrenergic signaling, during decision-making using a 2AFC task (see detailed description in 'Materials and methods'). Rats (female n = 10, male n = 9) were able to quickly learn the task and perform at stable levels (>100 trials, >70% accuracy for 3 d in a row, ~2 wk). The trial structure within our task is outlined in *Figure 1A*. Briefly, rats self-initiated trials by nosepoking in a central well and maintaining that nosepoke until one of two LED cues were illuminated in the back of the reward well (200–700 ms, randomized). The cue indicated which lever press would be rewarded on each trial and remained illuminated until the rat left the central nosepoke. Once behavioral performance was stable, animals were tested with the

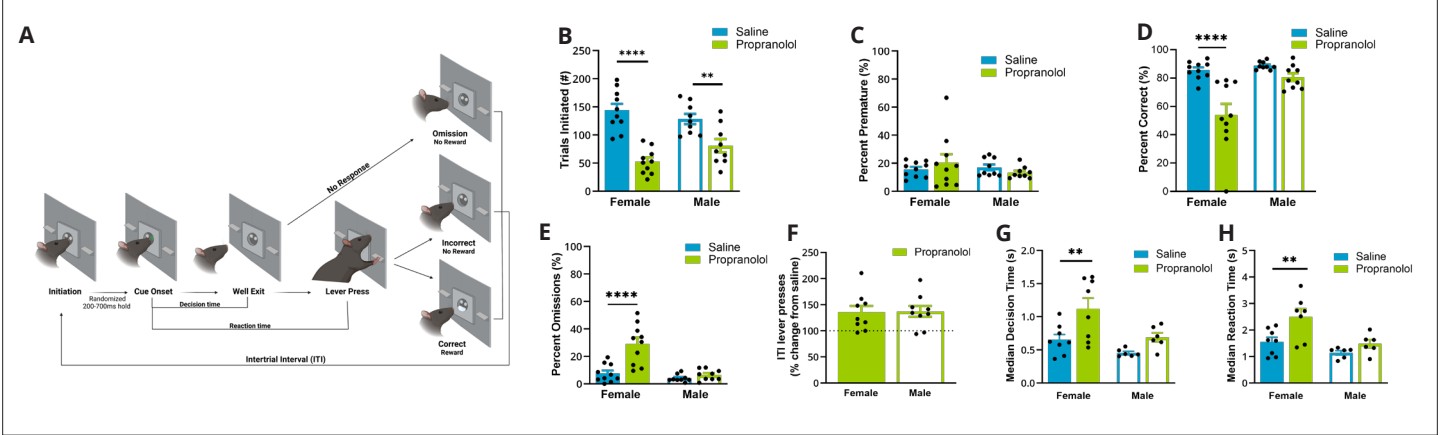

**Figure 1.** Propranolol decreased goal-directed task participation, particularly in females. (**A**) Schematic of events in the two-alternative forced-choice (2AFC) task (L–R). (1) Rats self-initiated trials by breaking an IR beam in the central well. (2) After a variable length (200–700 ms) IR beam hold, one of two randomly selected cues are illuminated that signals which lever would be rewarded when pressed. (3) Cues maintained illumination until rats left the well, 'Well Exit.' The time from cue onset (2) to well exit (3) was identified as the decision time. (4) Animals had 5 s from cue onset to execute a decision via lever pressing, 'Lever Press.' The time from cue presentation (2) to lever press (4) was identified as the reaction time. (5) Possible trial outcomes were correct (sucrose reward given), incorrect, omission (no lever press within 5 s), or premature (well exit prior to cue presentation). After each trial, there was an intertrial interval where the house light was illuminated, and new trials could not be initiated (incorrect and omissions: 10 s; correct: 5 s). Created with BioRender.com. (**B**) Propranolol (green) decreased the number of trials initiated during a 2AFC task in females (p<0.0001, solid bars) and males (p=0.0066, open bars) compared to saline (blue). (**C**) Propranolol did not alter the percent of trials that animals left the well during the variable hold prior to cue presentation (premature) in females or males. (**D**) Propranolol decreased overall accuracy in females (p<0.001) but not males (p=0.3336). (**E**) The percent of trials that a response was omitted significantly increased only in females after propranolol (p<0.0001). (**F**) Altered task performance was not a result of decreased arousal – motor output or lever pressing ability. The ratio of lever presses made during the intertrial interval (ITI) to the total lever presses (lever press index) increased in both sexes by 150% after propranolol. (**G**) Decision time, or the latency to behaviorally discriminate cues and select an action, measured from cue onset to well exit, differed by sex and propranolol. Propranolol significantly increased the median decision times in females (p=0.0071) but not males (p=0.3086). (**H**) Behaviorally, median reaction times, from cue onset to lever press, were significantly altered by propranolol and sex. Propranolol significantly increased median reaction times in females (p=0.0101) but not males (p=0.4922). * Indicates a significant effect compared to saline *p<0.05; **p<0.01; ***p<0.001; ****p<0.0001. Behavioral data (n=19; female n=10, male n=9) is presented as mean ± SEM after confirming normality (Shapiro-Wilk). Comparisons between groups were made with REML ANOVA and Sidaks post hoc tests.

The online version of this article includes the following source data for figure 1:

**Source data 1.** Two-alternative forced-choice (2AFC) behavior.

---

nonselective β-noradrenergic antagonist (propranolol, 10 mg/kg, IP) or saline while recording from anterior M2.

We found that propranolol decreased engagement in a 2AFC task in a sex-specific manner, impacting females more than males. We identified a reduction in the number of trials initiated after propranolol ($F_{(1,17)}$ = 52.31; p<0.0001; two-way ANOVA; *Figure 1B*) and an interaction between sex and propranolol ($F_{(1,17)}$ = 5.194; p=0.0359; two-way ANOVA). Propranolol decreased initiation in females by 63% from 144.00 ± 11.46 to 52.90 ± 7.14 (p<0.0001) trials. Propranolol decreased trial initiation to a lesser extent in males (37%) from an average of 128.33 ± 9.14 to 80.89 ± 11.85 trials (p=0.0066). Premature well exit prior to cue presentation was not affected by propranolol in either sex (means range from 13 to 26%; $F_{(1,17)}$ = 0.02224; p=0.8832; two-way ANOVA; *Figure 1C*). On trials where there was a cue presentation, accuracy was decreased after propranolol administration ($F_{(1,17)}$ = 24.12; p=0.0001; two-way ANOVA; *Figure 1D*) particularly in females (p<0.0001) with an effect of sex and treatment interaction ($F_{(1,17)}$ = 8.413; p=0.0100; two-way ANOVA). The change in accuracy after propranolol was driven by an increase in the percent of omitted trials in females (treatment × sex interaction $F_{(1,17)}$ = 14.92; p=0.0012; two-way ANOVA). Propranolol increased omissions in females fourfold from 7.73% ± 2.07% to 29.06% ± 4.59% (p<0.0001; *Figure 1E*). In males, omission rate was not significantly increased after propranolol 4.13% ± 0.87% vs. 6.57% ± 1.31% (p=0.7517).

It is possible that some changes in behavioral performance were a result of the nonselective or peripheral effects of β-noradrenergic antagonism that reduce activity across the board. We examined potential nonspecific motor changes by looking at lever presses during the intertrial interval (ITI).

Although propranolol decreased lever pressing during trials, it led to increased ITI lever presses in both sexes by approximately 150% ($F_{(1,16)}$ = 21.18; p=0.0003; REML ANOVA; *Figure 1F*). This increase in ITI presses after propranolol indicated that task-related actions, rather than general activity/lever pressing was primarily impacted by β-noradrenergic blockade.

In our task, animals could control cue duration by maintaining their nosepoke in the central port. As cue duration was self-controlled, we used this as a proxy for decision time. Propranolol increased median decision times (time from cue onset to well exit) ($F_{(1,24)}$ = 10.18; p=0.0039; two-way ANOVA; *Figure 1G*). Females in particular showed the largest increase in decision time after propranolol (p=0.0071; Sidak's multiple-comparisons test [MCT]). Propranolol almost doubled median decision time in females from 618.5 ms ± 79.3 ms to 1189 ms ± 162.3 ms and increased median decision time in males from 434.5 ms ± 23.23 ms to 676.1 ms ± 68.89 ms. The distribution of decision times reflected a rightward shift and increased variability in both sexes after propranolol (female p<0.0001; male p=0.0021; Brown–Forsythe test).

Reaction times were measured as the time from cue onset to lever press. This included the total decision time (duration of cue illumination) and the time it took rats to complete the required action. Behaviorally, median reaction times were significantly increased by propranolol ($F_{(1, 11)}$ = 10.01; p=0.009; REML ANOVA; *Figure 1H*). Post hoc tests determined that propranolol significantly delayed median reaction times from 1.55 s ± 0.17 s to 2.49 s ± 0.32s in females (p=0.0101) but did not alter median reaction times in males, from 1.14s ± 0.08s to 1.48 s ± 0.15s (p=0.4922). However, reaction time distribution increased variability after propranolol compared to saline in both sexes (female p<0.0001, male p<0.0001; Brown–Forsythe test).

## Propranolol influences the basal properties of female anterior M2 neurons

To understand how propranolol changes neural representation of action plans during 2AFC decision-making, we recorded neural activity from anterior M2 (A/P:+3 mm; M/L: ±1.2 mm; D/V: –1.5 mm). We collected 347 well-isolated single units (female n = 159, male n = 188) from deep layers of anterior M2 across 14 rats (female n = 8, male n = 6) after saline vehicle or propranolol (*Figure 2A and B*). Within a session, anterior M2 displayed heterogeneous event-related activity. We identified two primary task-related activity drivers: cue onset ('Cue') and lever press response ('Lever Press'). These events bookend the expected duration of action plan signals within anterior M2. Action plan signals in anterior M2 were well represented in both sexes. In females, the percent of anterior M2 units with task-locked modulation was 71.43% (n = 70) after saline and 44.26% (n = 27) after propranolol. In males, 66.36% (n = 71) of units demonstrated task-locked modulation after saline and 59.30% (n = 48) after propranolol. β-Noradrenergic signaling altered the proportion of anterior M2 neurons that showed task-locked activity in females but not males (female p=0.0008; male p=0.3604; Fisher's exact test). Only units with cue or lever press task-related activity were used for further analysis (see 'Materials and methods' for details on unit selection).

Across recording sessions, mean firing rates of task encoding neurons differed between sexes ($F_{(1,212)}$ = 6.387; p=0.0122; two-way ANOVA; *Figure 2C*), after saline but not propranolol (p=0.0061), with average activity if anterior M2 units being 3.40 ± 0.33 Hz in females vs. 5.25 ± 0.48 Hz in males after saline, and 4.51 Hz ± 0.54 Hz vs. 5.38 Hz ± 0.65 Hz after propranolol. In females, propranolol drove a rightward shift in the distribution of the neural firing rate in anterior M2 (p=0.0271; Mann–Whitney test; *Figure 2D*) indicative of reduced inhibitory tone on these neurons. We did not identify any impact on the distribution of firing rates recorded in males (p=0.8228; Mann–Whitney test; *Figure 2E*).

As previously noted, individual M2 neurons exhibit side preference/laterality encoding with increased firing for one cue and/or lever press side, and active inhibition for the opposite cue/lever press (*Li et al., 2016*; *Inagaki et al., 2018*; *Wei et al., 2019*). The proportion of neurons showing preferential excitation to either left or right cues was unaffected by propranolol in either sex (female p=0.6513; male p=0.5793; Fisher's exact test). However, the proportion of neurons showing preferential excitation with left/right lever press was altered by propranolol in both sexes (female p=0.0105; male p=0.0001; Fisher's exact test; *Figure 2F–I*), indicating a disruption of action-related signals.

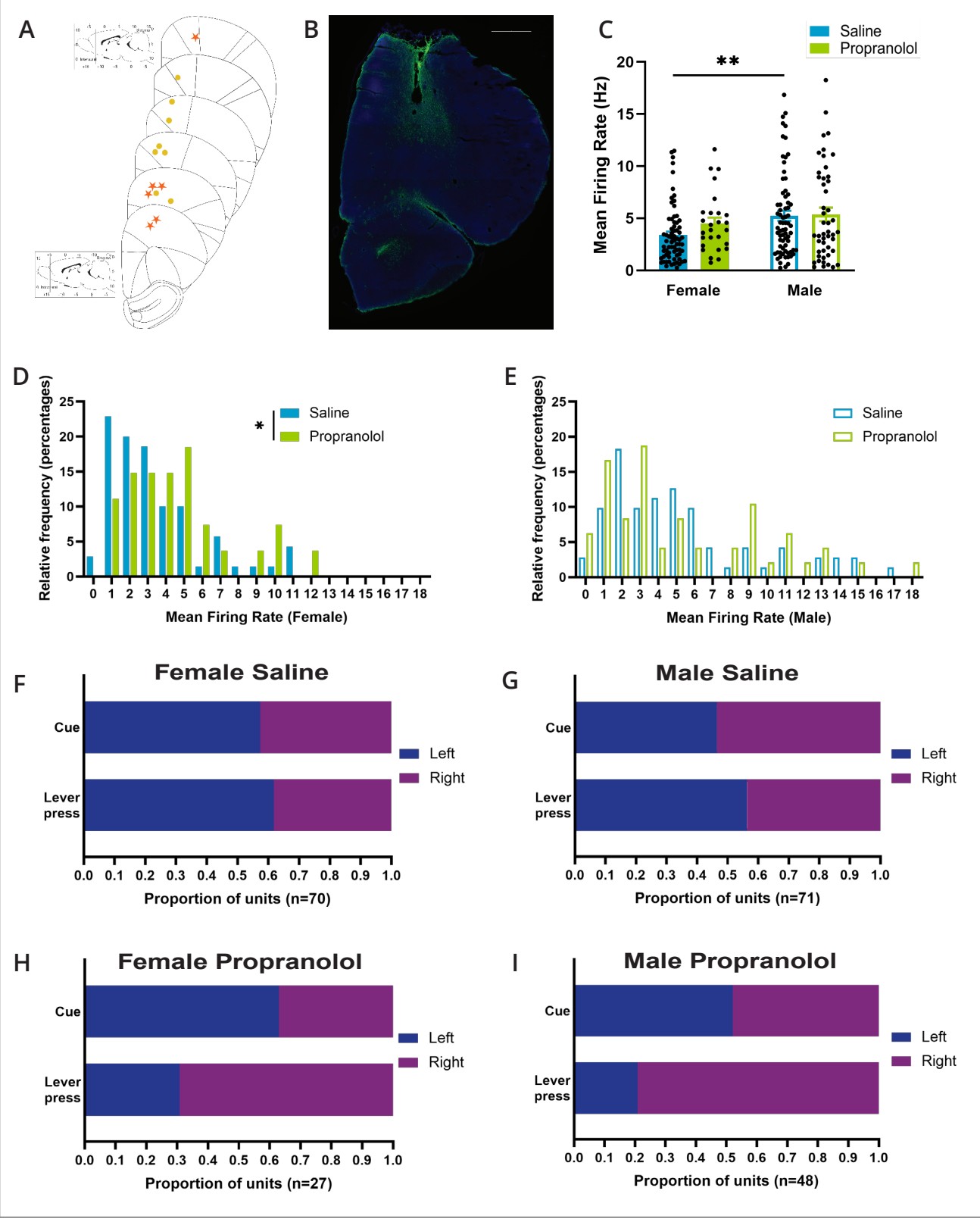

**Figure 2.** Propranolol influences basal properties of female anterior M2 neurons. (**A**) Placements of the center of electrode array in M2 in females (yellow circle) and males (orange star). (**B**) Representative figure of histological placement of microelectrode array in M2. Tissue is stained with DAPI (blue) and GFAP (green) to identify glial scarring after electrolytic lesions. Scale bar indicates 1000 μm. (**C**) Basal mean firing rates of anterior M2 neurons differed by sex (p=0.0061; Sidak's multiple comparison test). Propranolol did not alter mean firing rate in females or males (p=0.544; Sidak's multiple comparison

*Figure 2 continued*

test). Mean firing rate (Hz) of each unit included in electrophysiology is represented as individual points. Bar graphs represent the mean ± SEM of units from females and males after saline (blue; n=141, female n=70, male n=71) and propranolol (green; n=75, female n=27, male n=48). (**D**) The distribution of single-unit firing rate (Hz) in females was rightward shifted after propranolol (p=0.0271; Mann–Whitney test). Histograms show the distributions of firing rates in females after saline (blue) and propranolol (green) in 1 s bins where the value of the x-axis is the minimum value in each bin. (**E**) The distribution of single-unit median firing rate (Hz) in males was not significantly different after propranolol (p=0.8228; Mann–Whitney test). Histograms show the distributions of firing rates in males after saline (blue) and propranolol (green) in 1 s bins where the value of the x-axis is the minimum value in each bin. (**F-I**) Propranolol does not change side preference displayed in M2 neural activity around cue onset but does change the side preference for lever presses in females (p=0.0105; Fisher's exact test) and males (p=0.0001; Fisher's exact test). Proportions of units that display a left (blue) or right (purple) side preference around cue onset (top) and lever press (bottom) during the two-alternative forced-choice (2AFC) task separated by sex and treatment (F: female saline; G: male saline; H: female propranolol; I: male propranolol). Side preference was identified via a side preference index using the firing rate of neurons for 1 s after cue onset and 1 s prior to lever press (see 'Materials and methods'). Total number of units per treatment and sex is identified at the bottom of the graphs.

The online version of this article includes the following source data for figure 2:

**Source data 1.** M2 basal neural activity.

## β-Noradrenergic signaling is critical for suppression of irrelevant action plans in anterior M2

We further investigated how propranolol influenced encoding of action plans in anterior M2 on all trials in which a cue was received, including omitted trials and trials with completed actions (lever press). As expected, cue delivery led to opposing responses in individual neurons; neurons demonstrated strong excitatory responses for preferred cues and were actively inhibited (negative z-scores) after the presentation of nonpreferred cues (representative neuron, *Figure 3A*), suppressing irrelevant action plans.

We saw a reduction in action plan separation across anterior M2 neurons in both sexes after propranolol. Across the population, significant separation of opposing action plan signals (preferred and nonpreferred) in anterior M2 was sustained over 1.3 s in females and 1.1 s in males on saline sessions (*Figure 3B and C*, black lines indicate epochs of significant separation of action plans). When evaluating behavioral response events (*Figure 3B–E*, upper insets), sustained action plan separation in anterior M2 began after cue onset and largely continued until lever press responses. After propranolol, anterior M2 neurons in females did not significantly separate action plan signals at all (*Figure 3D*), and action plan separation was present for only short 100–300 ms bursts in males (*Figure 3E*). The loss of action plan separation in anterior M2 was primarily driven by propranolol acting directly or indirectly to disinhibit signaling from neurons after their nonpreferred cue. Neurons that would normally be silenced continued to show activity, introducing significant noise in anterior M2.

To verify whether anterior M2 encoded relevant task-related information (action plans for specific cues; right vs. left) after β-noradrenergic blockade, we compared population decoding accuracy of anterior M2 after saline and propranolol. As described in 'Materials and methods,' we trained a linear classifier to discriminate between neural activity patterns for left or right cues (*Meyers, 2013*). We then tested whether the classifier could use neural activity in each condition to accurately predict which cue had been delivered and whether this accuracy was above chance using shuffled data. We found that decoding was highly accurate and significantly above chance for sustained periods in saline conditions (*Figure 3F and G*); however, accuracy decreased in both female (*Figure 3H*) and male (*Figure 3I*) populations after propranolol and rarely rose above chance (solid bars on x-axis).

Overall, these data show that cue-evoked action plan firing within anterior M2 is disrupted after systemic β-noradrenergic blockade, with more severe impacts in females. Given the critical role for action plan signals in 2AFC behavior (*Inagaki et al., 2018*), loss of clearly separated action plans in anterior M2 likely contributed to behavioral deficits in our task, including increases in omitted trials (*Figure 1E*) and decision times (*Figure 1G*).

## Propranolol disrupts maintenance of action plans in anterior M2 leading up to actions

We next analyzed the data by aligning neural activity to lever press action on completed trials. Again, single units showed distinct side preferences for opposite lever press responses (representative neuron, *Figure 4A*).

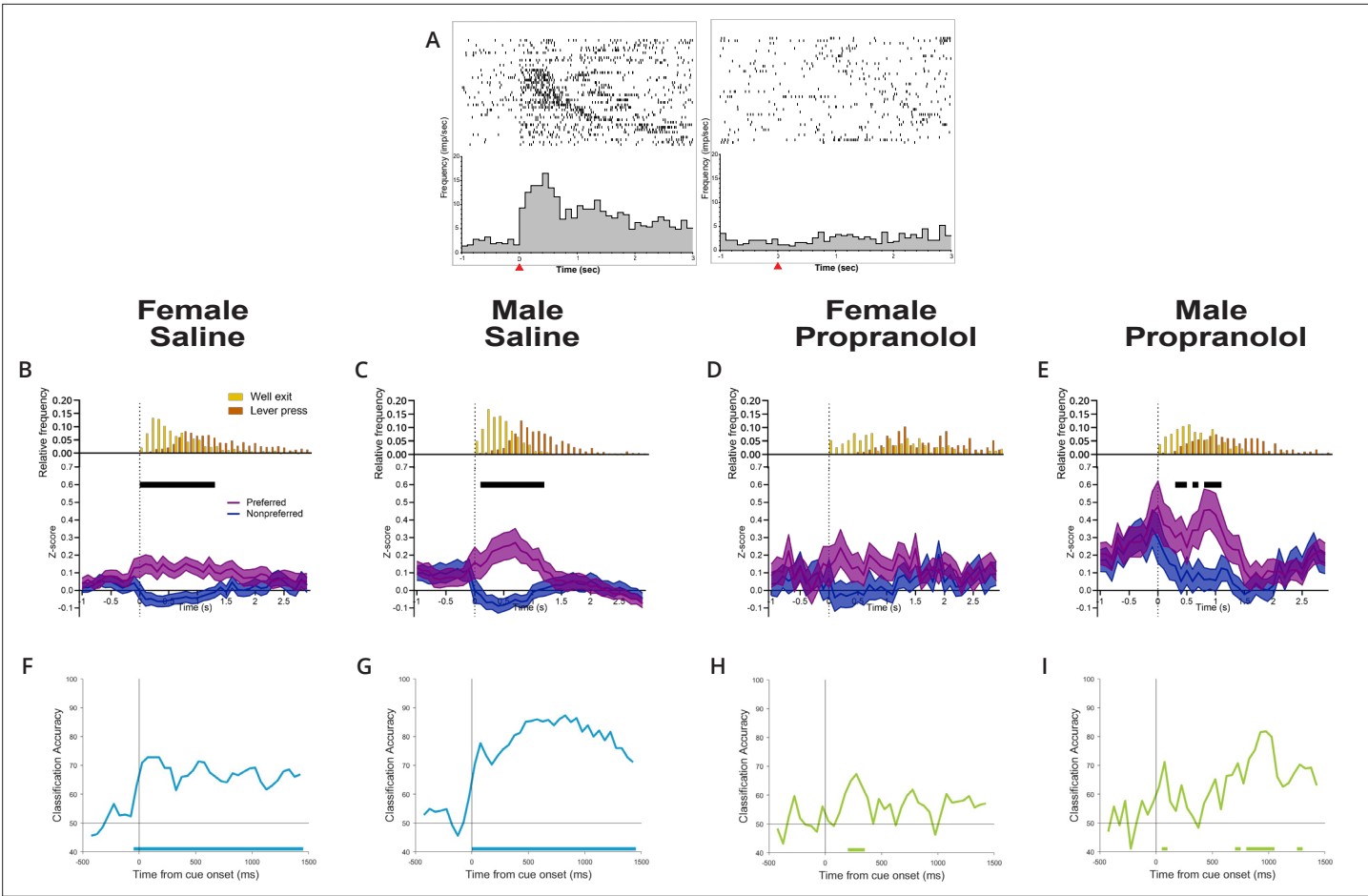

**Figure 3.** Propranolol decreases active inhibition of nonpreferred action plans. (**A**) Anterior M2 neurons show differential responses to opposing cue presentations. Raster (above) and perievent histogram of a single representative neuron aligned to cue onset (*t* = 0 s, 100 ms bins) from 1 s prior cue onset until 3 s after cue onset. The leftmost graph shows a single-unit response to left cue presentation, and the rightmost graph shows the same unit response to right cue presentation. This representative neuron was classified as left cue preferring, exemplified by a sharp and sustained increase in neural activity at left cue presentation but not right cue presentation. Red arrow on the x-axis identified cue onset at time = 0 s. (**B–E**) Propranolol decreases the inhibition of action plans for nonpreferred/irrelevant cues in females and males. Population neural data split by preferred (purple) and nonpreferred (blue) activity aligned to cue onset separated by sex and treatment (**B**: saline female, n=70; **C**: male saline, n=71; **D**: propranolol female, n=27; **E**: propranolol male, n=48). (**B, C**) Both sexes show task-related activity began shortly after cue onset that was sustained beyond cue offset at well exit (represented in histograms above Z-scored graphs, yellow) after saline. Neural activity on preferred (purple) and nonpreferred (blue) trials shows sharp and sustained increases and decreases, respectively, from cue onset until lever press. (**D**) Propranolol collapsed the separation of action plans between preferred and nonpreferred activity in females by decreasing task-related activity in both populations. (**E**) Propranolol blocked the active inhibition of nonpreferred action plans in males, increasing the noise represented in task-related activity. Graphs show mean ± SEM of z-scored activity on preferred (purple) and nonpreferred (blue) trials. Black bars above the graphs indicate significantly different population activity corrected for multiple comparisons (paired t-test with Holm-Sidak correction). Histograms above the z-scored population activity show the distribution of timepoints for well exit (yellow) and lever press (orange) relative to cue onset. (**F–I**) Propranolol reduced the performance of a linear classifier trained to predict cue identity (right/left) from M2 neural activity in males and females. Graphs show classification accuracy of decoding model trained with equal number of neurons (10) and trials (10/cue type) in 150 ms bins aligned to cue onset for each group (**F**: female saline; **G**: male saline; **H**: female propranolol; **I**: male propranolol). Classification accuracy for male and female saline (blue) groups was significantly above chance starting at cue onset and persisting for the remaining 1.5 s. Classification accuracy decreased in both female and male groups after propranolol (green) with almost no performance above chance. Bars on the bottom of each graph show the onset of time bins where the classification accuracy was significantly above chance.

The online version of this article includes the following source data for figure 3:

**Source data 1.** Cue-evoked M2 activity.

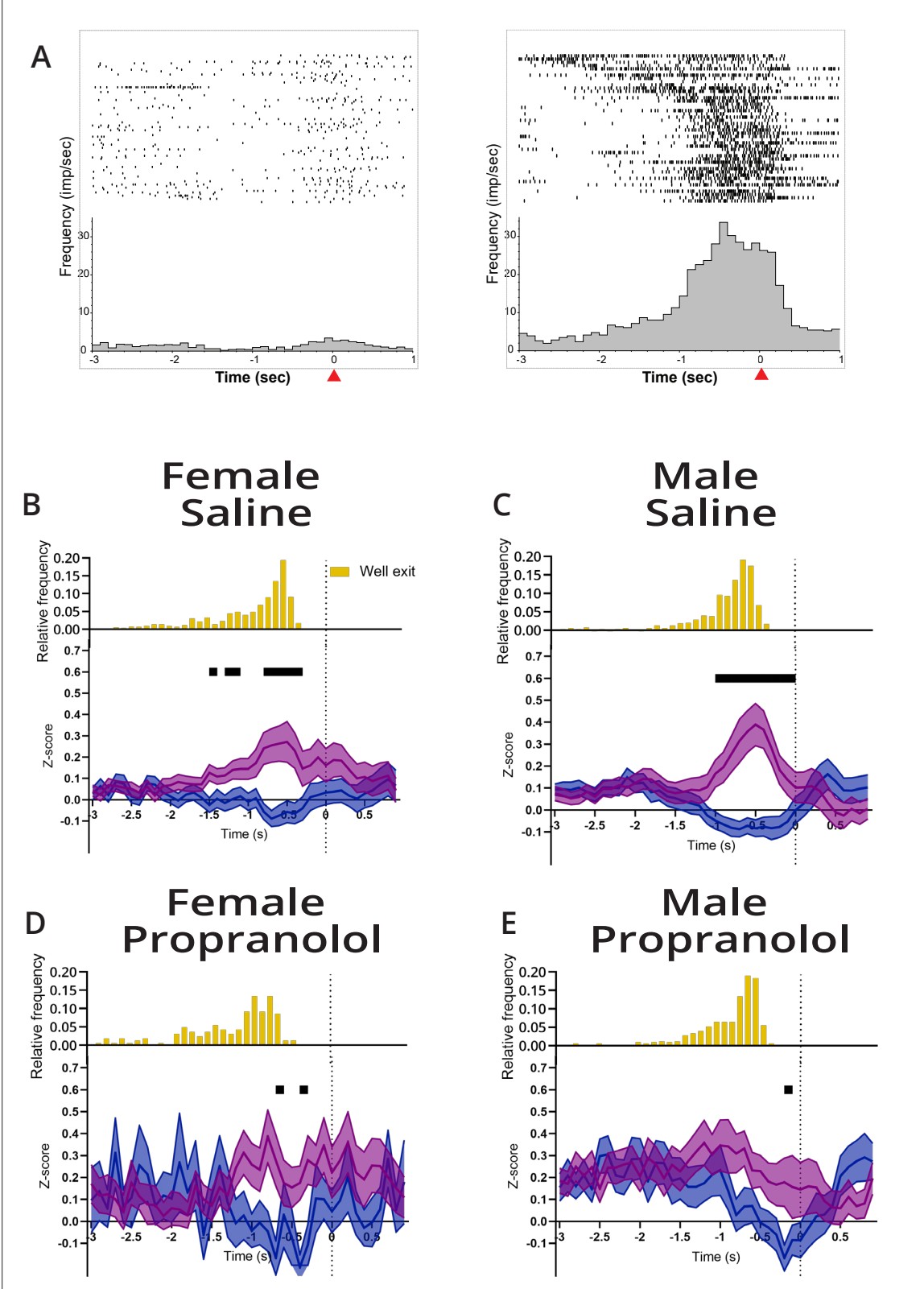

**Figure 4.** Propranolol decreases action plan representation prior to lever press. (**A**) Example differential responses to preferred and nonpreferred lever press. Raster (above) and perievent histogram aligned to lever press response (at *t* = 0 s, 100 ms bins) from 3 s before lever press to 1 s after lever press. Leftmost graph shows a single-unit response to a left lever press, and the rightmost graph shows the same unit response to a right lever press. This neuron was classified as right lever press preferring, exemplified by a sustained increased in neural activity leading up to the right lever press but not

*Figure 4 continued on next page*

*Figure 4 continued*

left lever press. Red arrow on the x-axis identified lever press at time = 0 s. (**B–E**) M2 represents action plans beginning at well exit that are sustained through to action execution (lever press). Propranolol decreased M2 action planning via decreased inhibition for nonpreferred lever and decreased excitation for the target (preferred) lever in both sexes. Population neural data split by preferred (purple) and nonpreferred (blue) lever press trials separated by sex and treatment (**B**: saline female, n=70; **C**: male saline, n=71; **D**: propranolol female, n=27; **E**: propranolol male, n=48). (**B, C**) Both sexes showed task-related activity began ~1 s prior to lever press and remained until action execution after saline. Neural activity on preferred and nonpreferred trials shows sharp and sustained increases and decreases, respectively, in activity prior to lever press execution. (**D**) Propranolol decreased the difference between preferred and nonpreferred activity in females. (**E**) Propranolol decreased the difference between preferred and nonpreferred activity in males. Graphs show mean ± SEM of z-scores, and black bars above the graphs show timepoints where the population activity of preferred and nonpreferred activity was significantly different (paired t-test with Holm-Sidak correction). Histograms above the z-scored population activity show the distribution of well exit times, respective to lever press (yellow).

The online version of this article includes the following source data for figure 4:

**Source data 1.** Lever press-aligned M2 activity.

Although action plans were identifiable in some neurons after propranolol, across anterior M2 sustained action plan representation was disrupted by β-noradrenergic blockade. Under vehicle conditions, action plans were represented in anterior M2 around 1 s before lever press (***Figure 4B and C***). Similar to the findings when trials were aligned to cue presentation, for completed actions propranolol disrupted sustained separation of action plans leading up to lever pressing in anterior M2 (***Figure 4D and E***). Inability to clearly separate action plans likely contributed to delayed reaction times after propranolol. Thus, the ability to select and maintain an action plan in anterior M2 appears to be greatly facilitated by β-noradrenergic signaling, whether that is β-noradrenergic signaling directly on M2 neurons or through a more indirect path. The disruption in the ability to maintain a selected action plan in M2 was most severe in females who also showed behaviorally the greatest increases in reaction times.

### Females show greater β2-noradrenergic receptor RNA density in M2

To determine whether biased noradrenergic signaling may contribute to the functional sex differences in propranolol's modulation of behavior and neural activity, we profiled noradrenergic receptor RNA expression within anterior M2 using RNAscope (***Figure 5A***). We quantified β1- and β2-noradrenergic receptor RNA as these subtypes are prominently expressed in frontal cortical regions (***Nicholas et al., 1991***; ***Rainbow et al., 1984***; ***Wanaka et al., 1989***). Further, propranolol is selective for β1 and β2 but not β3 receptors (***Baker, 2005***; ***Schena and Caplan, 2019***). We found that females had more cells containing β-noradrenergic receptor RNA in anterior M2 than males (effect of sex $F_{(1,14)}$ = 7.48; p=0.0161; two-way ANOVA; ***Figure 5B***), particularly β2-noradrenergic receptor RNA (p=0.01; Sidak's MCT). To determine whether β-noradrenergic receptor RNA expression was primarily found in pyramidal neurons or local interneurons, we colabeled for either excitatory (vGLUT1) or inhibitory (vGAT) amino acid transporter RNA. We found that much of the overall increase in β2-noradrenergic receptor RNA density in females was located on vGAT+ neurons within anterior M2 (effect of cell type $F_{(2,28)}$ = 7.02; p=0.0034; two-way ANOVA; ***Figure 5D***). β1-Noradrenergic receptor RNA was near exclusively found on vGLUT+ neurons in both sexes ($F_{(1,14)}$ = 225.5; p<0.0001; two-way ANOVA; ***Figure 5C***). This evidence provides one potential mechanism whereby decision-making in females critically relies on β2-noradrenergic modulation of local interneurons to suppress irrelevant information signaling in anterior M2 for efficient action planning.

## Discussion

How does norepinephrine impact the ability for anterior M2 neurons to encode action plans? Here we look at noradrenergic regulation of action planning by recording from anterior M2 neurons under β-noradrenergic blockade during a 2AFC decision-making task and characterizing β-noradrenergic receptor RNA expression in anterior M2. We found that blocking β-noradrenergic signaling blunted task-relevant neural activity. Propranolol, whether directly or indirectly, decreased the information encoded in anterior M2 during the 2AFC and behaviorally decreased task engagement and performance. The degree to which these behavioral and neural deficits occurred showed a functional sex difference where females displayed greater dependence on β-noradrenergic activity for action planning

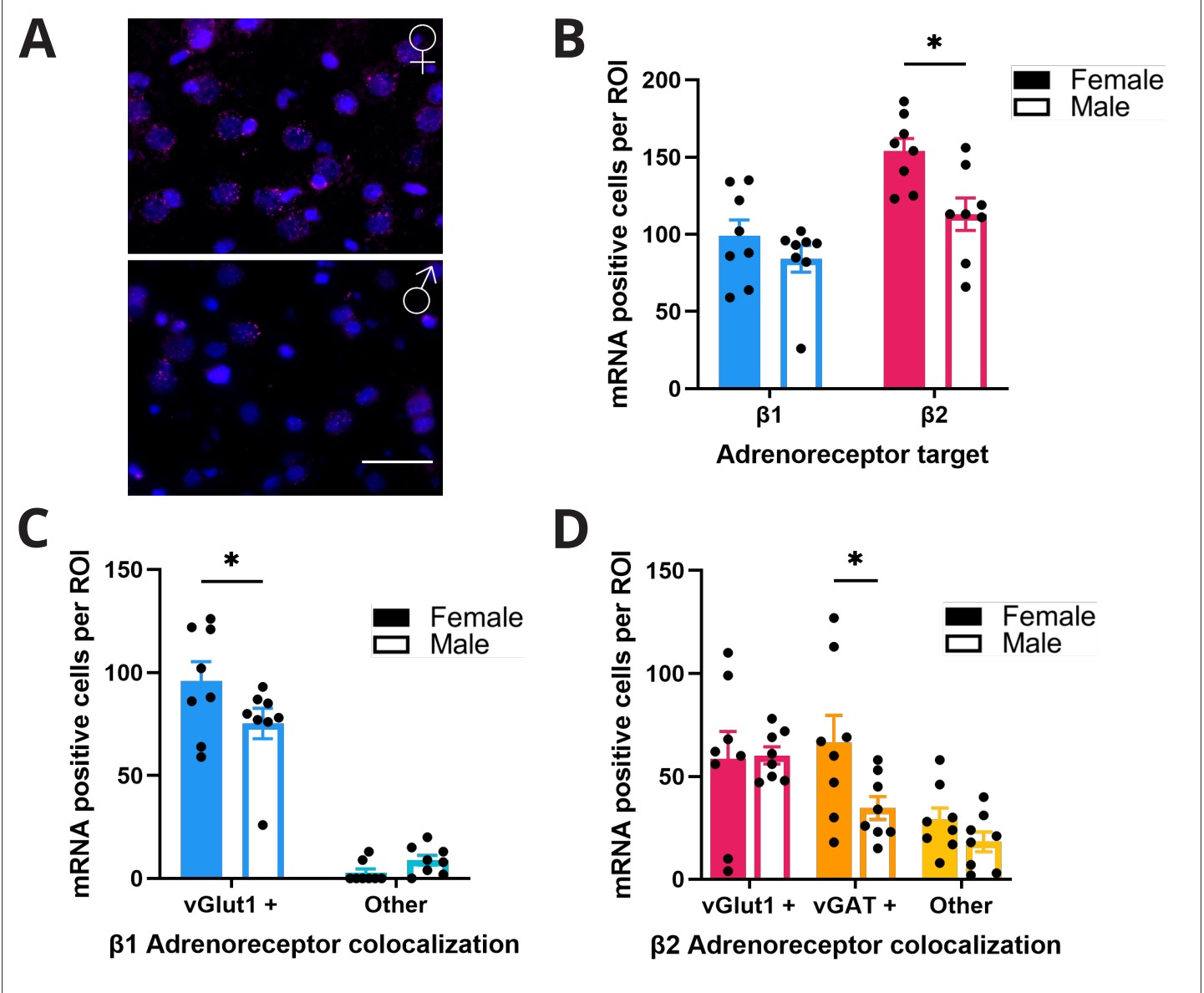

**Figure 5.** Females express higher levels of β-adrenoreceptor RNA in anterior M2. (**A**) Representative figure of RNAscope labeling in anterior M2 (n=8 hemispheres, 4 animals per group) of female (top) and male (bottom) rodents. β2 RNA expression (magenta) in M2 with DAPI (blue) counterstain. Scale bar indicates 50 μm. (**B**) Females have higher expression of β2 receptor RNA (pink) in M2 compared to males (p=0.01; Sidak's multiple comparisons test). There was no difference in β1 receptor RNA (blue) expression in females and males. (**C**) β1 receptor RNA expression was significantly higher on vGlut1+ cells (blue) in females compared to males (p=0.0453; Sidak's multiple comparisons test). There was no sex difference in the amount of β1 receptor RNA on non-vGlut1+ cells (light blue). (**D**) β2 receptor RNA expression was significantly higher on vGAT+ (orange) but not vGlut1+ (pink) or other (yellow) cells in females compared to males (p=0.0401; Sidak's multiple comparisons test).

The online version of this article includes the following source data for figure 5:

**Source data 1.** M2 RNAscope.

and performance. Females also showed higher levels of β-noradrenergic receptor RNA within anterior M2, particularly on local interneurons. Results from this study suggest that β-noradrenergic signaling is key to maintaining neural representation of action plans necessary for optimal performance during decision-making and provides evidence for sex differences within this mechanism.

The LC-NE system has been implicated in signal detection and altering gain of task-relevant cues in the cortex. NE acting amplifies salient cue-related signals while simultaneously quieting aberrant or 'noisy' background activity (*Clayton et al., 2004*; *Dayan and Yu, 2006*; *Servan-Schreiber et al., 1990*;

*Vazey et al., 2018*; *Waterhouse et al., 1998*). Our data confirms this role for NE in higher cognitive regions such as M2 and identifies that this mechanism can work specifically through β-noradrenergic receptors. The overall effect of blocking β-noradrenergic signaling, whether direct or indirect, was a reduction in the separation of preferred and nonpreferred action plans. Separation of action plans in anterior M2, which is normally sustained from cue onset through to the action, occurred briefly or not at all. Using a linear classifier, we showed that cue type could be decoded from anterior M2 neural activity after saline but decoding accuracy was lost after propranolol (*Figure 3F–I*). We would predict that diminished action plan encoding after propranolol would necessitate a longer period of time to complete actions or a failure to demonstrate behavioral choices. Indeed, this decreased separation of action plans, and reduced information within anterior M2, particularly in females, was associated with increases in trial omissions (*Figure 1E*), increased decision times (*Figure 1G*), and increased response times (*Figure 1H*). Collectively this evidence supports a strong role for noradrenergic signaling in efficient 2AFC decision-making. The role of norepinephrine is to promote neural gain and action plan representation within anterior M2 to optimize behavioral performance.

Previous studies have shown that M2 neurons display heterogeneous activity during decision-making tasks (*Chandrasekaran et al., 2017*; *Inagaki et al., 2018*; *Li et al., 2016*; *Svoboda and Li, 2018*). In particular, M2 units show preferential activation for cues/actions based on egocentric location and active inhibition for opposing cues/actions (*Inagaki et al., 2018*; *Wei et al., 2019*). Our data replicated these findings and show that these functions are consistent across sexes with similar task-related activity in anterior M2 neurons between females and males under saline. In this study, suppressing β-noradrenergic signaling altered baseline firing properties selectively in females and caused targeted deficits in neural representations of action plans. Our findings identify a specific role for global β-noradrenergic receptor activity in regulating the representation of action plans in anterior M2 during decision-making.

As M2 neurons have previously shown heterogeneous activity, it is possible that we captured neurons with different roles in decision-making, and by analyzing population activity, included neurons with ramping and discrete activity patterns. Previous studies have identified three patterns of activity in M2; firing after presentation of 'go' cue, ramping from cue onset to movement initiation, and firing during forced delay and movement (*Svoboda and Li, 2018*). While in our data ramping neural activity was not obvious when aligning the population activity to cue onset, when aligned to lever press ramping neural activity was observed. This suggests that our neural data is more tightly aligned to action planning, motor preparation, and execution than sensory integration, as previously reported. In this study by analyzing neural activity aligned to cue onset and lever pressing, we observed major effects of propranolol on the creation and maintenance of action plans even on trials where a behavioral response was omitted.

Baseline firing rates were increased after systemic propranolol in females only; task-related encoding in females was also more severely impacted than in males, highlighting a sex-specific vulnerability. Our data on differential RNA expression levels in anterior M2 provide support for propranolol acting directly within M2, with greater impact in females. There are known direct projections to M2 from the LC, the main supplier of NE to the forebrain. However, we must note that the behavioral and neural changes we observed are likely influenced by indirect and collective actions of propranolol in many regions.

Distinctions between the cognitive effects of β1 and β2 receptor activity in the brain have identified that β2 agonism improves working memory (*Ramos et al., 2008*) while β1 antagonism improves cognition (*Ramos et al., 2005*). Propranolol is a nonspecific β antagonist that has a high affinity for β1 and β2 but not β3 receptors (*Schena and Caplan, 2019*). Previous studies found that when administered systemically or intra-PFC, propranolol had no effect on a working memory task (*Arnsten and Goldman-Rakic, 1985*; *Li and Mei, 1994*), suggesting that the behavioral deficits found here are not due to working memory.

At baseline, animals perform high numbers of trials with high accuracy in part because 2AFC is a simple task, the animals are well trained, and the sucrose reward incentivizes high rates of performance. Propranolol significantly decreased trial initiations across the session. As the adaptive gain theory posits, aberrant levels of tonic NE release produce behavioral deficits with low tonic levels of NE resulting in behavioral disengagement (*Aston-Jones, 2004*; *Aston-Jones and Cohen, 2005*). Decreases in trial initiation after propranolol may be a result of animals not attending to the

environment and being disengaged. The reduction in trial initiations after propranolol may also reflect reduced motivation or arousal. Motivational influences were not directly tested in this study and are not commonly associated with propranolol. However, on saline session in which the trial rate decreases over the session we did not identify any difference in action plan signals within anterior M2 between early and late trials. We saw an increase in ITI lever pressing after propranolol across both sexes making arousal confounds unlikely in this study.

β-Noradrenergic receptors are found throughout the rat brain. In the cortex, both β1 and β2 receptors are heavily expressed while there is only limited β3 expression (*Nicholas et al., 1991*; *Rainbow et al., 1984*; *Schena and Caplan, 2019*; *Wanaka et al., 1989*). Within M2, we identified a sex difference in noradrenergic receptor RNA distribution with higher densities of β2 RNA in females (*Figure 5B*). Behavior and action plan signaling in anterior M2 was more severely impacted after propranolol in females than in males. Collectively these results may indicate a greater reliance on noradrenergic regulation of action planning in females. After propranolol, anterior M2 neurons had limited ability to create and maintain appropriate action plans, as seen by the reduced separation of preferred and nonpreferred activity in females (*Figures 3D and 4D*), also seen to a lesser degree in males (*Figures 3E and 4E*). Despite sex differences in β-noradrenergic RNA expression, males were not immune to the behavioral effects of altered neural activity after β receptor antagonism. Both males and females had rightward shifts in decision time distributions. Overall, these behavioral deficits were mediated by direct and indirect β-noradrenergic dependent alterations in decision-related activity in anterior M2. While both males and females display neural and behavioral deficits after propranolol administration, the larger impact on female behavioral performance may be attributed to greater neural disruption after propranolol. We propose that higher levels of β-noradrenergic receptor RNA support a greater dependence on β-adrenoreceptors to regulate neural activity in anterior M2.

Some aspects of decision-making strategies have known sex differences (*Chen et al., 2021a*; *Chen et al., 2021b*; *Orsini and Setlow, 2017*; *Shansky, 2018*), and the LC-NE system has known sex differences in its structure, size, and stress susceptibility (*Bangasser et al., 2011*; *Pinos et al., 2001*; *Valentino et al., 2012*). We have identified sex differences in decision-making and M2 function with regard to their sensitivity to perturbations of noradrenergic tone. We also identified a potential mechanism for this differential sensitivity by showing that within M2 there are distinct differences in β-noradrenergic receptor RNA expression density. Females showed a higher density of β-noradrenergic receptor RNA in M2, specifically β1 receptor RNA on glutamatergic neurons and β2 receptor RNA on GABAergic neurons (*Figure 5*). Prior evidence has demonstrated that females are more risk-averse in their decision-making (*Orsini and Setlow, 2017*). Having mechanisms for enhancing GABAergic tone in the decision-making regions of the brain would facilitate a risk-averse behavioral strategy by slowing decision times to allow greater information accumulation and reducing impulsivity. In this study, we found that the reduction in accuracy in females after propranolol was driven by an increase in omissions, rather than incorrect responses reflecting this risk-averse propensity. Noradrenergic regulation of local GABAergic interneurons in M2 appears critical for separating action plans during decision-making. Intact NE signaling increases the signal from neurons encoding cue-relevant information and suppresses activity from neurons that encode alternate irrelevant action plans. With increased β-noradrenergic receptor RNA expression in M2 of females, our results provide a potential biological basis for why disrupting NE would result in more pronounced physiological and behavioral outcomes in females. This highlights the potential for functional sex differences that emerge when neuromodulatory inputs are perturbed.

Action plans in M2 are established and maintained by promoting activity in some neurons while actively suppressing others. We found that blocking β-noradrenergic signaling led to a reduction in task-relevant encoding in anterior M2. The neural representation of opposing action plans in M2 showed reduced separation, decreased magnitude, and limited duration. This disruption in anterior M2 neural activity occurred alongside delayed and impaired decision-making as seen by increased decision times, reaction times, and omitted trials. We saw a sex bias of these impacts where females were more disrupted by blockade of β-noradrenergic signaling than males. We also saw that females had increased density of β-noradrenergic receptor RNA in M2 than males, providing a potential mechanism for this bias. Results from this study indicate that propranolol, by lowering β-noradrenergic signaling, can impair focused decision-making and task-related neural activity in anterior M2. Furthermore, sex-specific interactions in noradrenergic signaling influence cognitive function. The

measures of neural activity we observed indicate an important role of β-noradrenergic receptors in the maintenance and creation of an action plan during decision-making, specifically increasing the signal-to-noise of task-relevant neural representations. Future work should test the role of β receptor modulation explicitly within M2 neurons to test the direct influence on decision-making behavior.

## Materials and methods

### Animals

Long Evans rats (total n = 27; female n = 14, male n = 13; 6–8 weeks old; 200–275 g) were purchased from Charles River (006L/E, Wilmington, MA) and single housed on a reverse 12 hr light/dark cycle (lights ON at 9:00 PM). Separate cohorts of animals were used for in situ hybridization (female n = 4, male n = 4) and behavioral pharmacology/electrophysiology (female n = 10, male n = 9). Animals had access to ad libitum chow and water until behavioral testing began and were then restricted to 80% of ad libitum chow (female = 15 g, male = 20 g). Weight was monitored throughout the experiments to ensure maintenance of at least 80% free fed weight. All procedures were approved by the Institutional Animal Care and Use Committee at the University of Massachusetts Amherst in accordance with the guidelines described in the US National Institutes of Health *Guide for the Care and Use of Laboratory Animals* (National Research Council 2011). All animals were handled by an experimenter prior to electrode implant surgery and for 2 wk before behavioral training began. All animals were included in behavioral analysis. Of those animals, individuals with clearly identifiable single units that completed at least 20 trials during a recording session were included in electrophysiology analysis (n = 14).

### Custom static electrode array fabrication

HML VG bond-coated stainless-steel wire (25 µm diameter, California Fine Wire Company, Grover Beach, CA) was spun and bonded into stereotrodes with a Tetrodetwister (Open Ephys, Cambridge, MA). Each array was fabricated with 16 stereotrodes and 50 µm reference (1 mm above electrode tips) mounted to a nanostrip connector (Omnetics, Minneapolis, MN). A 32G insulated copper ground (Remington Industries, Johnsburg, IL) was connected to a stainless-steel screw in the occipital plate (McMaster Carr, Elmhurst, IL). A secondary 125 µm stainless-steel ground wire was placed between the dura and the skull over the parietal lobe contralateral to the implant. On the day of implant, stereotrodes were electroplated with a gold-noncyanide solution (50% diluted in saline, Sifco, Independence, OH) to achieve a target impedance of 0.1–0.2 MΩ.

### Surgery

For 2 d prior and 5 d post-surgery, animals had access to minocycline-treated water (100 mg/l, Ohm Laboratories, North Brunswick, NJ), weighed and replaced daily to maintain freshness and monitor water consumption. Minocycline has been shown to reduce inflammation and scarring of brain tissue after electrode implantation (*Rennaker et al., 2007*). To reduce respiratory secretions during surgery, animals were given atropine (0.04 mg, IP, Henry Schein Animal, Melville, NY) 10 min before anesthesia. Animals were anesthetized with vaporized isoflurane (4% induction,~2% maintenance) and administered the analgesic/anti-inflammatory Metacam (1 mg/kg, IP, Henry Schein Animal) and antibiotic cefazolin (0.33 mg, IM, McKesson, Irving, TX) prior to surgery. Animals' eyes were covered in ophthalmic ointment (Lubifresh P.M., Livonia, MI) to prevent drying, and the incision site was shaved and sterilized with iodine and 70% isopropyl alcohol. During surgery, body temperature was maintained with deltaphase thermal pads (Braintree Scientific, Braintree, MA). Animals were placed into a stereotaxic frame (David Kopf Instruments, Tujunga, CA) and the skull was leveled. A unilateral (randomized) craniotomy was made over anterior M2 (females right n = 5, left n = 3; males right n = 2, left n = 4). Four stainless steel 0/80 screws (McMaster Carr) were inserted into the skull to act as anchors for the implant. Electrode array was slowly lowered to the coordinates (A/P: +3 mm; M/L:±1.2 mm; D/V: –1.5 mm) at a rate of 0.2 mm/min. The craniotomy was packed with sterile Gelfoam (Pfizer, Kalamazoo, MI) and covered with dental cement (OrthoJet, Wheeling, IL). Metacam and cefazolin were administered 2 d postoperatively to reduce inflammation, pain, and risk of infection. Animals were monitored and allowed to fully recover for at least 2 wk post-surgery before beginning training.

### Behavioral task

Operant chambers (Med Associates, St Albans, VT) contained a central well with a recessed panel of LED cue lights and infrared (IR) entry beam, two laterally located levers, a speaker, and a house light.

A continuous fan and sound-attenuating chambers muffled outside noise. Behavioral data collection was controlled with MedPC IV (Med Associates). For all training and testing, rats performed for 15% sucrose reward (95 µl/trial) and animals could perform up to 250 trials in a session. All training and tests were performed during the animal's active cycle in a dimly red-lit room.

Two weeks after surgery, animals were food-restricted and trained on a 2AFC task where animals were shaped to associate spatially distinct LED cues with one lever for reward (*Figure 1A*). During the 2AFC task, rats self-initiated trials by breaking the IR entry beam to the central well during a 40 min session (max 250 trials). After maintaining the central well nosepoke for a variable hold period (200–700 ms, pseudorandomized), either a left or right LED cue light was illuminated (50% probability), which signaled the correct lever to press ('Cue onset'). The LED cue remained illuminated until rats withdrew from the well ('Well exit'). Rats could then press one of the two levers that remain extended throughout the task ('Lever press'). Premature withdrawal from the well before cue onset terminated a trial and was punished with a 10 s cued timeout. Correct responses on the cued lever produced 15% sucrose reward paired with a 100 ms 5 kHz tone followed by a 5 s ITI. Incorrect responses were punished with a 10 s cued timeout. Failure to respond on a lever within 5 s of withdrawal from the well counted as an omission and was punished with a 10 s timeout. ITIs and timeouts were indicated by house light illumination; extinguishing the house light signaled to rats that a new trial could be initiated. Rats were trained until their individual baseline performance was stable for at least 3 d based on total trials (±10%) and accuracy (>70% overall). When animals showed stable performance, electrophysiology recordings were obtained. On test days, 30 m before testing began rats were lightly and briefly anesthetized with isoflurane for head stage attachment and drug delivery.

## Drugs

The β-noradrenergic antagonist propranolol ((S)-(-)-propranolol [10 mg/kg], Tocris, Bristol, UK) was diluted in 0.9% saline to achieve a final concentration of 10 mg/ml. Drugs and vehicle (0.9% saline) were administered IP 30 m before testing, at a volume of 1 ml/kg. Drug order was randomized, and each test day was followed by 2–3 d of baseline performance. Rats underwent 1 d of propranolol treatment unless there was a problem with the electrophysiology recording during the session (n = 1).

## Data acquisition/electrophysiology recordings

Neural recordings were digitized at the headstage (HST/32D, Plexon; Dallas, TX), passed through a passive commutator and collected using an Omniplex data acquisition system (Plexon). Signals were sampled at 40 kHz, bandpass filtered 0.10 Hz to 7500 Hz, gain 1×, and stored offline for analysis. Recordings began 5 min before and ended 5 min after the task. MedPC timestamps were integrated through the Omniplex acquisition, and animal activity was observed and recorded during the task with a video camera during the session.

## Brain collection and electrode placement

At the end of behavioral experiments, animals were anesthetized with isoflurane and electrolytic current was passed through 4–6 wires per array (10 µA for 10 s) for post hoc verification of electrode position. Forty-eight hours later, animals were anesthetized with a ketamine xylazine solution (1.5 ml/kg, ketamine:xylazine 56:8.7 mg/ml) and transcardially perfused with ice-cold 0.9% saline followed by 4% paraformaldehyde. Immediately after, animals were perfused with 3% potassium ferrocyanide, 3% potassium ferricyanide, and 10% HCl solution for a Perl's reaction in iron deposits at the site of electrode tips. Brains were post-fixed in 4% paraformaldehyde overnight and cryoprotected in 20% sucrose azide prior to freezing and sectioning at 40 µm (CM3050S; Leica, Wetzlar, Germany). Tissue slices containing M2 were stained with neutral red and electrode placement was validated. Secondary validation of lesion placement was obtained by antibody staining for glial-fibrillary-associated protein (GFAP).

For GFAP staining, tissue sections were washed with PBS, blocked for 60 m in PBST and 3% NDS (Jackson Immuno Research, West Grove, PA), and incubated overnight on a shaker at room temperature in fresh blocking solution with polyclonal rabbit anti-GFAP (1:1000, Z0334, Dako, Santa Clara, CA). The following day tissue was washed in PBST and incubated for 2.5 hr at room temperature with Alexa Fluor 488 donkey anti-rabbit secondary (1:500, 711-545-152, Jackson ImmunoResearch). Tissue was counterstained with in DAPI (1:1000, 268298-10 mg, Millipore, Burlington, MA) in PBS for 10 min.

Tissue slices were washed in PBS, Tris buffer, mounted onto slides (Superfrost Plus, Fisher, Waltham, MA), covered with antifadant (Citifluor AF-1, Electron Microscopy Science, Hatfield, PA) and coverslipped. Tissue was imaged using a Zeiss Axio Imager M2 microscope (Zeiss, Oberkochen, Germany).

## In situ hybridization (RNAscope)

Animals were terminally anesthetized with a ketamine xylazine solution as above. Brains were removed, flash-frozen in cooled isopentane, and kept at –80°C until sectioning. Frontal blocks containing M2 were embedded in OCT compound (Sakura Finetek USA, Torrance, CA) and cryosectioned at 16 um onto Superfrost Plus slides (Fisher). Slide-mounted sections were processed using RNAscope Muliplex Fluorescent Reagent kit (Advanced Cell Diagnostics, Newark, CA) according to the manufacturer's protocol with probes targeted to β-noradrenergic receptors Rn-Adrb1 (468121) or Rn-Adrb2-C2 (468131-C2), and vesicular transporters for glutamate Rn-Slc17a7 (317001-C3) or GABA Rn-Slc32a1 (424541). Tissue was counterstained using DAPI and imaged using a Zeiss Axio Imager M2 microscope. DAPI counterstaining was used to identify anatomical landmarks and facilitate consistency in ROI acquisition. ROIs were taken from anterior M2 at the following coordinates: (a) AP +2.6 mm, ML ±1.0 mm, and DV 1.6 mm; (b) AP +3.0 mm, ML ±1.0 mm, and DV 1.8 mm; and (c) AP +3.4 mm, ML ±1.0 mm, and DV 2.0 mm. One M2 ROI was imaged at 20× from each hemisphere in all animals and used for manual quantification with FIJI by a blinded investigator (*Schindelin et al., 2012*). Data from each hemisphere was averaged across sections.

## Data processing, analysis, and visualization

Offline, cross-channel (>80%) artifacts were removed and a lowcut 250 Hz Bessel filter was applied. Single units (>4σ above background) were manually isolated using principal components analysis in Offline Sorter (Plexon). This resulted in a total of 347 well-isolated single units for analysis: 205 units (female n = 98, male n = 107) from 14 animals (female n = 8, male n = 6) after saline, and 142 units (female n = 61, male n = 81) across 11 animals (female n = 5, male n = 6) after 10 mg/kg propranolol.

Electrophysiology data was analyzed using Neuroexplorer (Plexon) and MATLAB (Natick, MA). We analyzed electrophysiology data around two periods of interest during the task: cue onset and lever press after cue presentation. M2 neurons have been shown to encode action selection from cue onset until decision execution in cue-driven tasks. We analyzed data aligned to both cue onset and lever press response to fully capture the complexity of M2 neural activity and to observe the neural data on trials where a cue was given but no lever press response was made (omitted trials).

To detect task-relevant neurons, we identified neurons with increased activity around either cue presentation or lever press response (saline: 141 out of 205 neurons; propranolol: 75 out of 142 neurons). To do this, we ran four paired *t*-tests per neuron comparing the number of spikes on each trial at specific task-related epochs (0 s to +1 s after right cue and left cue, and –1 s to 0 s before right lever press and left lever press). These epochs were compared to the number of spikes during 1 s in the middle of ITIs. p-Values were Bonferroni corrected for multiple comparisons. Electrophysiology analysis included all neurons with at least one significant corrected p-value during any of the four task epochs (saline = 141, female n = 70, male n = 71; propranolol = 75, female n = 27, male n = 48).

Units were then classified for side preference (a known feature of M2) at cue presentation and lever press response. Side preference was determined for each unit using a side preference index for both cue and lever press. Side preference index was calculated by comparing the number of spikes of each task epoch as noted above (1 s around cue/lever press) on left trials minus right trials divided by all trials (see equation below). A side preference index above 0 indicated a left preference and below 0 indicated a right preference. Side preference index was calculated for cue onset and lever press response for all included neurons.

$$\frac{\text{Spikes}_{\text{LeftCue}} - \text{Spikes}_{\text{RightCue}}}{\text{Spikes}_{\text{LeftCue}} + \text{Spikes}_{\text{RightCue}}}$$

For most neurons, the side preference was consistent for cue and lever press side (saline unchanged = 76.60%, changed = 23.40%; propranolol unchanged = 57.33%, changed = 42.67%). Overall, after saline administration 67 neurons were classified as right cue preferring (female n = 29, male n = 38) and 72 were left cue preferring (female n = 39, male n = 33). Around lever press after saline, 57 neurons were classified as right lever press preferring (female n = 26, male n = 31) and 82 were

classified as right lever press preferring (female n = 42, male n = 40). After propranolol administration, 33 were classified as right cue preferring (female n = 10, male n = 23) and 42 were left cue preferring (female n = 17, male n = 25). Around lever press after propranolol, 56 neurons were classified as right lever press preferring (female n = 18, male n = 38) and 18 were classified as right lever press preferring (female n = 8, male n = 10).

After classification, data from single units were z-scored in 100 ms bins normalized to its activity across the entire session. Z-scored activity was aligned to each task epoch (–1 s to 3 s aligned to cue onset, –3 s to 1 s aligned to lever press) and side of the cue and lever press (right/left; preferred/nonpreferred). Z-scored data is presented as the population mean ± SEM with trials separated by side preference, resulting in activity of each neuron represented in both the preferred and nonpreferred groups. For example, if a neuron has a left cue preference, the z-scored activity of that neuron on left cue presentation trials will contribute to the population mean of the preferred group, and the z-scored activity of that neuron on right cue presentation trials will contribute to the population mean of the nonpreferred group. Z-scored population activity between preferred and nonpreferred trials was compared using paired t-tests at each time bin. Multiple comparisons were corrected for using the Holm–Sidak method.

To compare decoding accuracy of anterior M2 neurons representing cue type after saline vs. propranolol treatment, we used a linear support vector machine to classify neural patterns around the emergence of action plans. We tested whether neural activity patterns in male and female M2 neurons accurately encode each cue type (right vs. left) after saline and propranolol using the Neural Decoding Toolbox (*Meyers, 2013*; *Rikhye et al., 2018*; *Zhang et al., 2011*). We sparsely trained the classifier to discriminate neural activity associated with left or right cues using a random sample of 10 presentations of each cue (right and left) from 10 random neurons from each population (saline female, saline male, propranolol female, propranolol male). Once trained, we tested the classifier on its ability to accurately predict/decode cue direction using neural data from all other trials in each condition. To evaluate whether decoding results were above chance, cue type labels were randomly shuffled and decoding analysis bootstrapped over 50 iterations creating a null distribution of chance decoding accuracy. Permutation tests were used to determine whether decoding accuracy within each group (saline female, saline male, propranolol female, propranolol male) was significantly different from the chance distribution of shuffled neural data from each group.

Behavioral and histological data was analyzed using GraphPad Prism v8 (San Diego, CA). Behavioral data is presented as mean ± SEM after confirming normality (Shapiro–Wilk) unless noted otherwise in the figure legends. Comparisons between groups were made with REML ANOVA and Sidak's post hoc tests. Distributions of firing rates were compared using Mann–Whitney tests. Distributions of decision times and reaction times were compared using Brown–Forsythe tests. Population z-scores on preferred and nonpreferred trials were compared using paired t-tests with Holm–Sidak multiple-comparisons correction. Graphs were compiled in Prism or MATLAB, and figures were compiled in Adobe Illustrator CS (San Jose, CA).

## Acknowledgements

The authors thank Michael Kelberman and Kara Conlan for their help in training animals.

## Additional information

### Funding

| Funder | Grant reference number | Author |
| --- | --- | --- |
| National Institute of Mental Health | R00MH104716 | Elena M Vazey |
| National Institute of Mental Health | F31MH131348 | Ellen M Rodberg |

The funders had no role in study design, data collection and interpretation, or the decision to submit the work for publication.

## Author contributions
Ellen M Rodberg, Conceptualization, Software, Formal analysis, Investigation, Visualization, Methodology, Writing - original draft, Writing - review and editing; Carolina R den Hartog, Investigation; Emma S Dauster, Formal analysis, Investigation, Visualization, Methodology; Elena M Vazey, Conceptualization, Resources, Supervision, Funding acquisition, Visualization, Methodology, Project administration, Writing - review and editing

## Author ORCIDs
Ellen M Rodberg (ID) http://orcid.org/0000-0002-4857-3970
Elena M Vazey (ID) http://orcid.org/0000-0003-3311-9414

## Ethics
All procedures were approved by the Institutional Animal Care and Use Committee at the University of Massachusetts Amherst (#2018-0080) in accordance with the guidelines described in the US National Institutes of Health Guide for the Care and Use of Laboratory Animals (National Research Council 2011). All surgery was performed under isoflurane anesthesia, and every effort was made to minimize suffering.

## Decision letter and Author response
Decision letter https://doi.org/10.7554/eLife.85590.sa1
Author response https://doi.org/10.7554/eLife.85590.sa2

# Additional files

## Supplementary files
• MDAR checklist

## Data availability
Data analyzed during this study are included in the supporting file.

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
