## [Editor Report]

This study provides valuable evidence of a potential sex-dependent, biological mechanism for propranolol-driven changes in decision-making. Though the functional component directly linking premotor cortex activity to performance is incomplete, the discovery of a molecular bias in adrenoreceptor expression that may underlie its impact on behavior is novel and compelling. These experiments present a useful framework for future inquiry linking direct actions within premotor cortex for decision-making changes in both males and females.

---

## [Decision Letter]

**Decision letter after peer review:**

Thank you for submitting your article "Sex-dependent noradrenergic modulation of premotor cortex during decision making" for consideration by *eLife*. Your article has been reviewed by 2 peer reviewers, and the evaluation has been overseen by a Reviewing Editor and Michael Frank as the Senior Editor. The following individual involved in the review of your submission has agreed to reveal their identity: Sebastian Bouret (Reviewer #2).

Essential revisions:

1) There is a lack of specificity due to the systemic antagonism of β adrenergic via propranolol. Thus, changes in M2 firing rates could be due to indirect activity changes elsewhere. This makes it impossible to conclude that changes in behavior are due to direct effects of activity within M2. Authors should either provide new data with local infusions in this region or attenuate their claims.

2) Authors need to better describe the task and behavior. Specifically, the contribution of M2 to this task and whether the changes in M2 activity observed are causally related to the behavioral disruptions are not clear. For example, it is possible this task engages M2 in motivation, not cue-guided (more cognitive aspects of) behavior.

3) Rationale and interpretation of the sex differences observed should be expanded upon. Authors should better describe if there is a functional sex difference (due to receptor distribution), or if the drug has distinct effects on males vs females.

4) The description and analyses of M2 neural activity are superficial and should be expanded on.

*Reviewer #1 (Recommendations for the authors):*

– The number of propanol treatment days is unclear, and whether they were intermixed with saline days is not stated.

– Should be noted that this is anterior M2 in Results, giving coordinates, as there are varied results across the large AP swath of M2.

– The axis and legend numbers on the PETH graphs need to be increased in size to be legible.

– Not clear whether p values calculated in Figure 3 F-I, as well as throughout the manuscript in regards to neural activity analyses, were corrected for the number of comparisons made. An uncorrected p-test presentation does not seem to be appropriate here.

– Not clear is how consistency in the counting of positive cells in M2 slices was achieved, or the location of the slice where cells were counted.

– Care should be taken in interpreting the results, specifically in the Results section. For example, instead of referring to non-goal-directed and goal-directed lever presses (lines 112, 116, 117), the wording should adhere more closely to what was actually measured (lever presses during ITI or during cued). There were no tests performed to examine whether lever pressing during ITI was goal-directed or not (i.e. outcome devaluation or contingency degradation procedures).

– Check consistency between norepinephrine and noradrenergic language.

---

## [Author Response]

Essential revisions:1) There is a lack of specificity due to the systemic antagonism of β adrenergic via propranolol. Thus, changes in M2 firing rates could be due to indirect activity changes elsewhere. This makes it impossible to conclude that changes in behavior are due to direct effects of activity within M2. Authors should either provide new data with local infusions in this region or attenuate their claims.

Alternate mechanisms are now indicated more clearly, and we have attenuated definitive claims of direct actions throughout the manuscript.

We agree that we cannot directly conclude from these experiments that the behavioral and neural changes are a result of propranolol acting solely on anterior M2, as opposed to acting on M2 in addition to other regions that may influence the behavior or physiology of anterior M2 or our task. The systemic nature of our propranolol manipulations enables many opportunities for indirect or direct modulation. Our studies collectively describe the constellation of changes after propranolol administration across both sexes and functional impacts on cognitive performance. We have used use supporting evidence from our experiments using mRNA expression, and existing literature on behavioral functions of M2 (Barthas and Kwan, 2017; Gremel and Costa, 2013; Guo et al., 2017; Inagaki et al., 2018, 2022; Li et al., 2016; Siniscalchi et al., 2016; Sul et al., 2011; Wei et al., 2019), sex differences in behavioral strategies (Grissom and Reyes, 2019) and central noradrenergic systems (Bangasser et al., 2011, 2016; Luque et al., 1992; Mulvey et al., 2018; Ohm et al., 1997; Pinos et al., 2001) to propose a potential biological mechanism for understanding propranolol driven changes. Namely the proposed mechanism highlights the possibility of direct actions within anterior M2 which is supported by multiple lines of data in our study but does not exclude other mechanisms. We explicitly state in both the results and discussion that the deficits in anterior M2 neural activity may be a result of propranolol direct action or an indirect due to propranolol acting on other brain regions.

2) Authors need to better describe the task and behavior. Specifically, the contribution of M2 to this task and whether the changes in M2 activity observed are causally related to the behavioral disruptions are not clear. For example, it is possible this task engages M2 in motivation, not cue-guided (more cognitive aspects of) behavior.

We have elaborated on our specific task and behavioral measures in the methods section. In the Results section we summarize the main principles of behavior we are evaluating. We also now include a more detailed infographic to clarify the task for all readers that is presented at the beginning of the Results section (Figure 1A). In addition, we have more clearly described and referenced the known role of M2 in 2AFC performance.

We apologize for any confusion regarding our task design, and for not clearly establishing the known roles of M2 in action planning within 2AFC tasks. The causal role of M2 in cue-guided 2AFC behaviors is well documented (Barthas and Kwan, 2017; Gremel and Costa, 2013; Guo et al., 2017; Inagaki et al., 2018, 2022; Li et al., 2016; Siniscalchi et al., 2016; Sul et al., 2011; Wei et al., 2019). M2 is necessary for accurate performance in 2AFC. Bilateral photo-inhibition of M2 between cue presentation and behavioral response (or subsets within that epoch) reduces 2AFC performance to chance levels in highly trained animals (Guo et al., 2017; Inagaki et al., 2018; Li et al., 2016). We have made sure to highlight this relevant background and clarify M2 function.

This does not exclude the potential motivational influence of propranolol on 2AFC or M2. M2 activity (Giamundo et al., 2021) has recently been implicated in motivation for different size rewards, however in our task reward volumes are consistent across the session. All trials in our task are self-initiated and require a random hold duration (200-700ms) prior to cue delivery, requiring motivation to initiate trials. Behaviorally we see a large decrease in task engagement (number of trials initiated through required hold duration), although trials are evenly distributed across the sessions after propranolol. This decrease in engagement reflects predicted noradrenergic mediated adaptive gain including direct or indirect influence on motivation.

We investigated whether motivational changes may be reflected in our anterior M2 recordings. Representative PETHs from the first and last 10 trials within a session did not identify differences in anterior M2 activity, although as noted above trial rate is consistent across propranolol sessions. During saline sessions the trial rate decreases across the session in both sexes, potentially reflecting a slight reduction in motivation with increasing satiety. However neural activity remains consistent between the first and last 10 trials, reflective of cue-guided action plans across the session (Author response image 1).

**Author response image 1. sa2fig1:** Motivation, as measured by the number of self-initiated trials in the 2AFC task decreased over time after saline but neural representation of action plans did not. Top: The number of self-initiated trials binned in 5-minute increments and plotted across the 40-minute session. Males (open circle) and females (closed circle) start with high rates of trial initiation, ~25trials/5min, after saline (left) which decreases over the session. This may reflect motivational changes as satiety increases over time. Trial initiations are consistent, ~10 trials/5min, throughout propranolol sessions (right) in males and females. Bottom: Although motivation to initiate trials decreased from the beginning to end of the session, the neural representations of action plans in anterior M2 did not change in females (left) or males (right). After preferred (pink) and nonpreferred (blue) cue onset, neurons show the same patterned activity during the first 10 trials (bright pink/blue) and the last 10 trials (dark pink/blue) suggesting the decreased motivation to initiate trials under saline (top figure) is not reflected in the M2 population activity.

3) Rationale and interpretation of the sex differences observed should be expanded upon. Authors should better describe if there is a functional sex difference (due to receptor distribution), or if the drug has distinct effects on males vs females.

We agree this is an interesting finding from our results and we now clarify and expand on this interpretation in our discussion.

In the current study we did not identify evidence for a distinct action of propranolol across sexes. Qualitatively, behavioral, and physiological changes were similar between sexes, however females showed significantly larger changes after β adrenoreceptor blockade, highlighting a functional sex difference when the system was perturbed/challenged using propranolol. We identified one potential mechanism underlying that functional difference with relevant differences in adrenoreceptor mRNA density. Our results combined with the existing literature suggest there is potential for sex differences in physiology and behavior due to functional differences in noradrenergic system organization between males vs females. However, we cannot conclusively determine that the functional sex differences we saw in our results are only due to noradrenergic actions within anterior M2, other mechanisms and other brain regions likely contribute.

4) The description and analyses of M2 neural activity are superficial and should be expanded on.

More detail describing the neural analyses has been added to the methods section.

Reviewer #1 (Recommendations for the authors):– The number of propanol treatment days is unclear, and whether they were intermixed with saline days is not stated.

We apologize for this oversight and now provide additional detail in our methods section. Recordings were taken from a single treatment day and this is now noted in the methodology under drugs. In rare cases that technical difficulties with the electrophysiology acquisition arose (n=1) due to insecure cables, partial recordings were discarded, and an additional treatment day was repeated at a later date. Treatment order of propranolol/saline days was random and there were 2-3 days of baseline behavioral performance with (no treatment or recording) in between each treatment day.

– Should be noted that this is anterior M2 in Results, giving coordinates, as there are varied results across the large AP swath of M2.

We agree that M2 has a broad longitudinal axis. We have provided more details, including specific coordinates for our M2 targeting in our Results section in addition to prior reference in the methods. For added clarity we have changed reference to M2 to “anterior M2” throughout the revised manuscript.

– The axis and legend numbers on the PETH graphs need to be increased in size to be legible.

We thank the reviewer for identifying this and have increased the axis and legend numbers on the PETH graphs.

– Not clear whether p values calculated in Figure 3 F-I, as well as throughout the manuscript in regards to neural activity analyses, were corrected for the number of comparisons made. An uncorrected p-test presentation does not seem to be appropriate here.

To improve rigor and clarity we have removed the panels from Figure 3 of sliding window analyses representing uncorrected p values. We have revised our criterion for identifying significantly modulated neurons and classifying their side preference by focusing on four relevant task epochs – (1) left and (2) right cue presentation, (3) left and (4) right lever press. We have retained only neurons showing activity significantly above that seen in ITI periods in at least one of those epochs using Bonferroni corrected pvalues. These revised criteria changed the absolute number of units identified as having significant task related activity from 292 (first submission) to 216 (revised submission). We have revised all relevant results and figures so that only neurons conforming to the new criteria (n=216) are included. Although this altered specific values for many analyses of neural firing, our findings, overall statistical results (at the appropriate α values) and conclusions remain the same. We have confirmed that all analyses have been corrected for multiple comparisons as appropriate and include these revised selection criteria details in the methods section.

– Not clear is how consistency in the counting of positive cells in M2 slices was achieved, or the location of the slice where cells were counted.

We have provided additional description in our methods section to clarify how assessment of mRNA expression was quantified. Specifically, DAPI counterstaining was used to identify anatomical landmarks in each hemisphere across three sections from each animal. Images were taken from anterior M2 at the following coordinates (a) AP +2.6mm, ML +/- 1.0mm and DV 1.6mm, (b) AP +3.0mm, ML +/- 1.0mm and DV 1.8mm and (c) AP +3.4mm, ML +/- 1.0mm and DV 2.0mm These locational details are now included in our methods. Counting of these images was undertaken by an investigator blinded to the location the image was taken from and the sex of the subject.

– Care should be taken in interpreting the results, specifically in the Results section. For example, instead of referring to non-goal-directed and goal-directed lever presses (lines 112, 116, 117), the wording should adhere more closely to what was actually measured (lever presses during ITI or during cued). There were no tests performed to examine whether lever pressing during ITI was goal-directed or not (i.e. outcome devaluation or contingency degradation procedures).

We appreciate the attention to using specific language for actions within our task. We have replaced generalized wording including references to goal-directed lever presses with more specific language, specifically ITI lever presses vs “in trial” lever presses in both the text and figures.

– Check consistency between norepinephrine and noradrenergic language.

We have carefully reviewed our manuscript for references to norepinephrine and noradrenaline. We use the americanized term “norepinephrine (NE)” to refer to the neurotransmitter alone. We use global term noradrenergic to refer to the signaling network including receptor targets. We have resolved any instances to conform with this convention.

References cited

Aston-Jones, G., and Bloom, F. (1981). Nonrepinephrine-containing locus coeruleus neurons in behaving rats exhibit pronounced responses to non-noxious environmental stimuli. *The Journal of Neuroscience*, *1*(8), 887–900. https://doi.org/10.1523/JNEUROSCI.01-08-00887.1981

Bangasser, D. A., Wiersielis, K. R., and Khantsis, S. (2016). Sex differences in the locus coeruleusnorepinephrine system and its regulation by stress. Brain Research, 1641, 177–188. https://doi.org/10.1016/j.brainres.2015.11.021

Bangasser, D. A., Zhang, X., Garachh, V., Hanhauser, E., and Valentino, R. J. (2011). Sexual dimorphism in locus coeruleus dendritic morphology: A structural basis for sex differences in emotional arousal. Physiology and Behavior, 103(3–4), 342–351. https://doi.org/10.1016/j.physbeh.2011.02.037

Barthas, F., and Kwan, A. C. (2017). Secondary Motor Cortex: Where Sensory Meets Motor in the Rodent Frontal Cortex. Cell Press, 40(3), 181–193. https://doi.org/10.1016/j.tins.2016.11.006

Clayton, E. C., Rajkowski, J., Cohen, J. D., and Aston-Jones, G. (2004). Phasic activation of monkey locus ceruleus neurons by simple decisions in a forced-choice task. Journal of Neuroscience, 24(44), 9914–9920. https://doi.org/10.1523/JNEUROSCI.2446-04.2004

Daie, K., Svoboda, K., and Druckmann, S. (2021). Targeted photostimulation uncovers circuit motifs supporting short-term memory. Nature Neuroscience, 24(2), 259–265. https://doi.org/10.1038/s41593-020-00776-3

Giamundo, M., Giarrocco, F., Brunamonti, E., Fabbrini, F., Pani, P., and Ferraina, S. (2021). Neuronal Activity in the Premotor Cortex of Monkeys Reflects Both Cue Salience and Motivation for Action Generation and Inhibition. The Journal of Neuroscience, 41(36), 7591–7606. https://doi.org/10.1523/jneurosci.0641-20.2021

Giordano, N., Alia, C., Fruzzetti, L., Pasquini, M., Palla, G., Mazzoni, A., Micera, S., Fogassi, L., Bonini, L., and Caleo, M. (2023). Fast-Spiking Interneurons of the Premotor Cortex Contribute to Initiation and Execution of Spontaneous Actions. The Journal of Neuroscience, 43(23), 4234–4250. https://doi.org/10.1523/JNEUROSCI.0750-22.2023

Gremel, C. M., and Costa, R. M. (2013). Premotor cortex is critical for goal-directed actions. Frontiers in Computational Neuroscience, 7(110). https://doi.org/10.3389/fncom.2013.00110

Grissom, N. M., and Reyes, T. M. (2019). Let’s call the whole thing off: evaluating gender and sex differences in executive function. Neuropsychopharmacology, 44(1), 86–96. https://doi.org/10.1038/s41386-018-0179-5

Guo, Z. V., Inagaki, H. K., Daie, K., Druckmann, S., Gerfen, C. R., and Svoboda, K. (2017). Maintenance of persistent activity in a frontal thalamocortical loop. Nature, 545(7653), 181–186. https://doi.org/10.1038/nature22324

Inagaki, H. K., Chen, S., Daie, K., Finkelstein, A., Fontolan, L., Romani, S., and Svoboda, K. (2022). Annual Review of Neuroscience Neural Algorithms and Circuits for Motor Planning. https://doi.org/10.1146/annurev-neuro-092021

Inagaki, H. K., Inagaki, M., Romani, S., and Svoboda, K. (2018). Low-dimensional and monotonic preparatory activity in mouse anterior lateral motor cortex. Journal of Neuroscience, 38(17), 4163– 4185. https://doi.org/10.1523/JNEUROSCI.3152-17.2018

Li, N., Daie, K., Svoboda, K., and Druckmann, S. (2016). Robust neuronal dynamics in premotor cortex during motor planning. Nature, 532, 459–464. https://doi.org/10.1038/nature17643

Luque, J. M., De Bias, M. R., Segovia, S., and Guillam6n, A. (1992). Sexual dimorphism of the dopaminebeta-hydroxylase-immunoreactive neurons in the rat locus ceruleus. In Developmental Brain Research (Vol. 67).

Mulvey, B., Bhatti, D. L., Gyawali, S., Lake, A. M., Kriaucionis, S., Ford, C. P., Bruchas, M. R., Heintz, N., and Dougherty, J. D. (2018). Molecular and Functional Sex Differences of Noradrenergic Neurons in the Mouse Locus Coeruleus. Cell Reports, 23(8), 2225–2235. https://doi.org/10.1016/j.celrep.2018.04.054

Ohm, T. G., Busch, C., and Bohl, J. (1997). Unbiased estimation of neuronal numbers in the human nucleus coeruleus during aging. Neurobiology of Aging, 18(4), 393–399. https://doi.org/10.1016/S0197-4580(97)00034-1

Pinos, H., Collado, P., Rodríguez-Zafra, M., Rodríguez, C., Segovia, S., and Guillamón, A. (2001). The development of sex differences in the locus coeruleus of the rat. Brain Research Bulletin, 56(1), 73–78. https://doi.org/10.1016/S0361-9230(01)00540-8

Rajkowski, J., Majczynski, H., Clayton, E., and Aston-Jones, G. (2004). Activation of monkey locus coeruleus neurons varies with difficulty and performance in a target detection task. Journal of Neurophysiology, 92(1), 361–371. https://doi.org/10.1152/jn.00673.2003

Ramos, B. P., and Arnsten, A. F. T. T. (2007). Adrenergic pharmacology and cognition: Focus on the prefrontal cortex. Pharmacology and Therapeutics, 113(3), 523–536. https://doi.org/10.1016/j.pharmthera.2006.11.006

Siniscalchi, M. J., Phoumthipphavong, V., Ali, F., Lozano, M., and Kwan, A. C. (2016). Fast and slow transitions in frontal ensemble activity during flexible sensorimotor behavior. Nature Neuroscience, 19(9), 1234–1242. https://doi.org/10.1038/nn.4342

Sul, J. H., Jo, S., Lee, D., and Jung, M. W. (2011). Role of rodent secondary motor cortex in value-based action selection. Nature Neuroscience, 14(9), 1202–1210. https://doi.org/10.1038/nn.2881

Vazey, E. M., Moorman, D. E., and Aston-Jones, G. (2018). Phasic locus coeruleus activity regulates cortical encoding of salience information. Proceedings of the National Academy of Sciences of the United States of America, 115(40), E9439–E9448. https://doi.org/10.1073/pnas.1803716115

Wang, M., Ramos, B. P., Paspalas, C. D., Shu, Y., Simen, A., Duque, A., Vijayraghavan, S., Brennan, A., Dudley, A., Nou, E., Mazer, J. A., McCormick, D. A., and Arnsten, A. F. T. (2007). α2A-Adrenoceptors Strengthen Working Memory Networks by Inhibiting cAMP-HCN Channel Signaling in Prefrontal Cortex. Cell, 129(2), 397–410. https://doi.org/10.1016/j.cell.2007.03.015

Wei, Z., Inagaki, H., Li, N., Svoboda, K., and Druckmann, S. (2019). An orderly single-trial organization of population dynamics in premotor cortex predicts behavioral variability. Nature Communications, 10(216). https://doi.org/10.1038/s41467-018-08141-6